# Maternal Fever and Reduced Fetal Movement as Predictive Risk Factors for Adverse Neonatal Outcome in Cases of Congenital SARS-CoV-2 Infection: A Meta-Analysis of Individual Participant Data from Case Reports and Case Series

**DOI:** 10.3390/v15071615

**Published:** 2023-07-24

**Authors:** Elena S. Bernad, Florentina Duica, Panagiotis Antoniadis, Andreea Moza, Diana Lungeanu, Marius Craina, Brenda C. Bernad, Edida Maghet, Ingrid-Andrada Vasilache, Anca Laura Maghiari, Diana-Aurora Arnautu, Daniela Iacob

**Affiliations:** 1Department of Obstetrics and Gynecology, Faculty of Medicine, “Victor Babes” University of Medicine and Pharmacy, 300041 Timisoara, Romania; bernad.elena@umft.ro (E.S.B.); craina.marius@umft.ro (M.C.); iacob.daniela@umft.ro (D.I.); 2Clinic of Obstetrics and Gynecology, “Pius Brinzeu” County Clinical Emergency Hospital, 300723 Timisoara, Romania; 3Center for Laparoscopy, Laparoscopic Surgery and In Vitro Fertilization, “Victor Babes” University of Medicine and Pharmacy, 300041 Timisoara, Romania; 4Bucharest Emergency Clinical Hospital, 014461 Bucharest, Romania; florentina.duica80@gmail.com; 5Alessandrescu-Rusescu National Institute for Mother and Child Health, Fetal Medicine Excellence Research Center, 020395 Bucharest, Romania; 6Department of Biochemistry and Molecular Biology, Faculty of Science, University of Southern Denmark, 5230 Odense, Denmark; panosant89@gmail.com; 7Center for Modeling Biological Systems and Data Analysis, “Victor Babes” University of Medicine and Pharmacy, 300041 Timisoara, Romania; dlungeanu@umft.ro; 8Department of Functional Sciences, Faculty of Medicine, “Victor Babes” University of Medicine and Pharmacy, 300041 Timisoara, Romania; 9Department of Neuroscience, Faculty of Medicine, “Victor Babes” University of Medicine and Pharmacy, 300041 Timisoara, Romania; bernad.brenda@umft.ro; 10Center for Neuropsychology and Behavioral Medicine, “Victor Babes” University of Medicine and Pharmacy, 300041 Timisoara, Romania; 111st Department, Faculty of Dental Medicine, “Victor Babes” University of Medicine and Pharmacy, 300070 Timisoara, Romania; edidamaghet@gmail.com; 12Department of Obstetrics and Gynecology, Faculty of Medicine, “Grigore T. Popa” University of Medicine and Pharmacy, 700115 Iasi, Romania; tanasaingrid@yahoo.com; 131st Department, Faculty of Medicine, “Victor Babes” University of Medicine and Pharmacy, 300041 Timisoara, Romania; boscuanca@yahoo.com; 14Department of Cardiology, Faculty of Medicine, “Victor Babes” University of Medicine and Pharmacy, 300041 Timisoara, Romania; aurora.bordejevic@umft.ro

**Keywords:** SARS-CoV-2, congenital, vertical transmission, pregnancy, newborn

## Abstract

Objectives: To determine risk factors for primary and secondary adverse neonatal outcomes in newborns with congenital SARS-CoV-2 infection. Data sources: PubMed/MEDLINE and Google Scholar from January 2020 to January 2022. Study eligibility criteria: newborns delivered after 24 weeks of gestation with confirmed/possible congenital SARS-CoV-2 infection, according to standard classification criteria. Methods: Execution of the IPD analyses followed the PRISMA-IPD statement. Univariate non-parametric tests compared numerical data distributions. Fisher’s exact or Chi-square test determined categorical variables’ statistical significance. Multivariate logistic regression revealed risk factors for adverse neonatal outcome. Results: Maternal fever was associated with symptomatic congenital infection (OR: 4.55, 95% CI: 1.33–15.57). Two-thirds of women that reported decreased fetal movements were diagnosed with IUFD (*p*-value = 0.001). Reduced fetal movement increased the risk of intrauterine fetal death by 7.84 times (*p*-value = 0.016, 95% CI: 2.23–27.5). The risk of stillbirth decreased with gestational age at the time of maternal infection (*p*-value < 0.05, OR: 0.87, 95% CI: 0.79–0.97). Conclusions: Maternal fever and perception of reduced fetal movement may be predictive risk factors for adverse pregnancy outcome in cases with congenital SARS-CoV-2 infection.

## 1. Introduction 

The current pandemic of the coronavirus disease 2019 (COVID-19), caused by SARS-CoV-2 (Severe Acute Respiratory Syndrome Coronavirus 2), has reached an unprecedented level of severity, which has culminated in a global public health crisis [1]. Despite the significant progress made in a concise amount of time in terms of diagnosis, treatment, and preventionof the SARS-CoV-2 infection in adults, pregnant women, and children, the infection of the fetus is still not completely understood [2,3,4].

Studies concerning vertical transmission of the SARS-CoV-2 virus are still at an early stage. While the physiopathological mechanism of SARS-CoV-2 transmission from the mother to the fetus during pregnancy is full of controversial hypotheses, reported histological evidence on the presence of the SARS-CoV-2 at the level of the placenta and in stillborn fetuses cannot be overlooked [4,5].

The WHO (World Health Organization) and Shah et al. have proposed various criteria for diagnosing vertical transmission, however since the diagnosis protocol is multistep and time-framed in both situations, the literature is scarce when seeking research that applied the proper tools for diagnosing vertical transmission [6,7]. Although some papers have found no evidence of SARS-CoV-2 vertical transmission, extensive studies report a rate of 1.8% to 5.3%. However, none of them mentions by which standard vertical transmissionwas diagnosed [8,9,10,11,12,13,14,15,16]. In a retrospective analysis of 145 pregnant infected women who delivered in Italy over five months, Di Guardo (2021) found a 5% vertical transmission rate using Shah’s categorization system [17]. The subject of potential risk factors for mother-to-fetus viral transmission remains unexplored.

In general, fetal infection can occur via a variety of pathways. While some viruses pass transplacentally and use endocytosis and transcytosis to avoid the placental immune system, other infections provoke direct disruption of the maternal-fetal interface [18,19]. Nonetheless, in certain instances, the ascending route, defined as the transfer of infected cells or the virus from the cervicovaginal compartment, has been discovered as a possible transmission route [20].

Some viral infections, such as congenital cytomegalovirus (CMV) disease, could be symptomatic and even life-threatening. In contrast, others can go unnoticed at birth but with potential long-term consequences on the infant [21].

The neonatal outcome varies depending on the type of viral infection; however, certain maternal, obstetrical, and neonatal factors might influence infant short-term as well as long-term outcome. Primary maternal CMV infection, for example, was more likely to be associated with symptomatic disease in the newborn [22]. Maternal clinical status at the time of infection and the timing of maternal infection throughout pregnancy seem relevant in determining whether or not poor neonatal outcome occurs. For instance, infants that develop sequela following CMV infection in utero are more likely to have been exposed to the virus in the first trimester of pregnancy [21]. In certain circumstances, such as active Herpes Simplex Virus-2 (HSV-2), the period between maternal infection and birth may have a significant influence on newborn well-being since the likelihood of HSV-2 transfer to the fetus is estimated to be 25% to 60%, ifmaternal infection occurs around the time of delivery [23]. Obstetrical variables such as prolonged rupture of membranes and mode of delivery are also essential in determining whether or not congenital infection is manifested in infants in the case of viruses with genital shedding [24].

From this perspective, in the case of the SARS-CoV-2 virus, current epidemiological studies have demonstrated that most newborns from infected mothers show no signs of infection. However, a small proportion are born symptomatic [25]. Despite this, the correlation between the newborn’s infective status and in-utero infection was not studied. A more concerning aspect is the higher rate of intrauterine fetal demise (IUFD) in case of maternal infection; its causes are still not thoroughly studied [26]. Although still premature, some studies claim long-term neurodevelopmental and ophthalmologic sequelae in infants born from mothers infected with the coronavirus during pregnancy [27,28].

The main focus of this study is to determine potential risk factors for adverse neonatal outcome in case of vertical transmission of the coronavirus virus.

## 2. Methods

The database created for a previous scoping review following PRISMA (Preferred Reporting Items for Systematic Reviews and Meta-Analyses) recommendations was rebuilt and updated with reported cases of confirmed and possible congenital SARS-CoV-2 disease, published until 30 January 2023 [29,30]. The database and statistical analysis followed the PRISMA-IPD guidelines [31].

### 2.1. The Search Strategy

The investigation continued to look for published cases in whom intrauterine and intrapartum coronavirus exposure was diagnosed using standard criteria. Search engines (PubMed/MEDLINE and Google Scholar) and MESH keyword (‘COVID-19*’ OR ‘SARS-CoV-2*’) AND (‘vertical transmission’ OR ‘in-utero transmission’ OR ‘congenital transmission’ OR ‘placental infection’) remained the same; however, we took into consideration all publications irrespective of language, as opposed to the previous article. Ultimately, eligibility criteria remained the same, except that only cases with viable pregnancies were included (24 weeks as opposed to 20 weeks of gestation in the previous review). The viability threshold was established according to the international poll of maternal-fetal medicine professionals [32]. An additional 210 new articles were found, of which 9 articles were included in the study (Figure 1).

### 2.2. Updated Eligibility Criteria for the Meta-Analysis

The following inclusion criteria were used to determine the eligible articles:Implementation of WHO or Shah’s guidelines to diagnose intra-uterine and intrapartum exposure to SARS-CoV-2 infection [6,7].Delivery after 24 weeks of gestation;Application of stringent infection control and prevention measures;Mother-neonate separation after delivery.

### 2.3. Definition of Congenital SARS-CoV-2 Infection Following Standard Criteria

Adapted tables incorporating both classification systems were used to establish the diagnosis of congenital SARS-CoV-2 disease. The WHO classification primed over the Shah’s classification system.

Irrespective of the used classification system, one should confirm in utero exposure and viral persistence in case of livebirths. Each recommendation classified vertical transmission as confirmed, possible, unlikely and indeterminate. There is a slight difference between the two recommendations. Table 1 and Table 2 describe the algorithms used for cases classified as confirmed or possible vertical transmission.

The difference between confirmed and possible cases of congenital SARS-CoV-2 infection stands from the sampling method. For instance, if the persistence of infection is demonstrated through sterile sampling (neonatal blood collected between 24–48 h after birth), in-utero exposure is confirmed (Table 1). However, if viral persistence is demonstrated through non-sterile sampling (nasopharyngeal swab), vertical transmission is classified only as possible.

In case of stillbirths, the WHO classification system considers an infection confirmed solely if there is positive fetal tissue sampling [7]. However, if fetal tissue sampling is not accessible, but there is positive placenta or amniotic fluid, vertical transmission is categorized only as possible, in contrast to Shah’s categorization, which considers such situations to be confirmatory as well [6,7]. All the stillbirths in the lot were classified using the WHO classification system (Table 3).

### 2.4. Outcome

Primary neonatal adverse outcome was defined as stillbirth or presence of SARS-CoV-2 symptoms in the newborn (in the first 24 h). Secondary neonatal adverse outcome was defined as neonatal death due to COVID-19 disease and presence of sequela of COVID-19 disease in the neonate at the moment of discharge from the hospital.

### 2.5. Quality Assessment

After updating the database and excluding unviable pregnancies, there were 55 case series/reports and three descriptive studies. Case series and case reports were assessed using the Scoring Criteria proposed by Murad [33]. A mean score of 3.09 (fair quality) resulted. In clinical studies (3), the Newcastle–Ottawa Scale was used to assess the quality, resulting in a mean of 5.3 (medium quality) [34]. In each of the 3 studies, a detailed description of the infected neonate was reported. Appendix A describes complete information and scoring for all the included articles.Each study had individual-level data, albeit several patient characteristics had more missing data than others. By excluding patients with complete data, 60% of the sample would have been eliminated, possibly introducing selection and reporting biases. Due to this aspect, we offer complete data.

### 2.6. Extraction of Relevant Data

The 58 articles detailed the outcomes of 82 pregnant women with confirmed SARS-CoV-2 infection and their 85 newborns (35 were confirmed, and the rest were classified only as possible congenital infection). Table 1, Table 2 and Table 3 describe all the criteria used for diagnosing congenital SARS-CoV-2 infection in this study.

Next, an IPD database was built using Microsoft Excel (version 2020, MS Corp., Redmond, WA, USA) that contained: details about the article, maternal and neonatal characteristics. Maternal metrics included socio-demographic features, associated comorbidities, clinical status related to COVID-19 symptoms and eventual obstetrical symptoms that urged the patient to the hospital. We paid particular attention to gestational age (GA) at the time of infection and at the time of birth, as well as the time lapse between maternal infection and pregnancy outcome (ΔT) and mode of delivery. Maternal vaccination status was also investigated; however due to scars number of women with known vaccination status (6/82), this metric was not included in the statistical evaluation.

Neonatal metrics included: pregnancy outcome (live birth/stillbirth), type of delivery (for live births) and other newborn characteristics (5-min Apgar Score, birthweight and its assessment in centiles -in accordance to the Nicolaides fetal and neonatal population chart, newborn sex, presence of any symptoms that could raise the suspicion of congenital SARS-CoV-2 infection), neonatal evolution during hospitalization, as well as presence of SARS-CoV-2 related sequela at the moment of neonatal discharge [35].

### 2.7. Special Consideration Regarding the Timelapse between Maternal Infection and Pregnancy Outcome

The method employed to determine the time lapsed (ΔT- measured in days) between the commencement of infection (T0) and the outcome of the pregnancy(T1) was different depending on the outcome of pregnancy and the mother’s condition. For asymptomatic pregnant patients, the first day of the count began on the day when a positive maternal PCR-SARS-CoV-2 test was acquired (Figure 1). T0 was the first day when symptoms manifested in symptomatic individuals. In the event of live births, the last day (T1) of the count was the day of delivery, whereas, in the case of stillbirths, it was the day when IUFD was confirmed (Figure 2).

### 2.8. Data Analysis

The observed frequency counts (%) were included in the descriptive statistics for the categorical variables, and the medianInter Quartile Range (IQR) was used for numerical variables. The descriptive table also contained the mean and the standard deviation for maternal and gestational age. Not all variables followed a normal distribution (Shapiro-Wilk statistical test was employed for checking the normality of the distributions). A comparison of the distribution of numerical data across two groups, where applicable, was carried out using univariate non-parametric statistical tests (ANOVA, Mann-Whitney U). The asymptotic, Fisher’s exact, Chi-square test, or Kruskal-Wallis statistical test was used to determine whether or not there was a statistically significant correlation between the categorical variables. Odds risk were evaluated using logistic regression analysis.

The statistical analysis was carried out with a degree of confidence equal to 95% and a level of statistical significance equal to 5%. Every probability value that was presented used a two-tailed test.

The statistical analysis was carried out using DataTab [36].

## 3. Results

Of the included 58 studies, nine were carried out in the United States [37,38,39,40,41,42,43,44,45], six in Italy [46,47,48,49,50,51], five in Brazil [52,53,54,55,56], four in France [57,58,59,60], four in Iran [61,62,63,64], three in Germany [65,66,67], three in Spain [68,69,70], two in Portugal [71,72], two in Romania [73,74], two in Russia [75,76], one in Peru [77], Slovakia [78], Columbia [79], China [80], Turkey [81], Switzerland [82], Ireland [83], India [84], Canada [85], UK [86], Mexico [87], Singapore [88], Malaysia [89], Belgium [90], Japan [91], Georgia [92], Saudi-Arabia [93], Sweden [94].

### 3.1. Baseline Maternal and Neonatal Characteristics

Among the 82included pregnant women, 59 (71.95%) had signs suggestive for COVID-19 disease at T0, whereas 52 women (63.41%) were symptomaticat T1. Overall, fever was the most reported symptom (54.8%), followed by respiratory (47.56%), neurologic (12.2%) and flu-like symptoms (15.85%).

The median maternal age was 30 in the symptomatic group and 30.5 in the asymptomatic group.

Regarding the severity of COVID-19 disease, 8 (9.76%) women were classified as having severe pneumonia, 2 of which required ventilatory and hemodynamic support. In these patients, the disease progressed rapidly, ΔT had a mean of 8 days (IQR 6–11 days) as opposed to patients that had mild disease or were asymptomatic, where the mean was 12.4 (ranging from 4–12 days) (*p*-value = 0.94).

Forty women were admitted to the hospital due to pregnancy-related issues. In 23% of cases, they reported reduced fetal movement, 17.07% had painful uterine contractions, 7.32% presented vaginal bleeding, and in 2.44%, premature rupture of membranes was diagnosed.

A total of 85 newborns with possible and confirmed congenital SARS-CoV-2 infection were included in the study. Only 28% (24) remained asymptomatic during hospitalizations, while the rest (72%) either had COVID-19 symptoms (24) or were stillborn (37). Most symptomatic newborns had respiratory symptoms (cough, episodes of apnea and tachypnea- 11 cases were diagnosed with pneumonia), followed by fever, digestive symptoms (feeding difficulties and vomiting and abdominal distension) and neurologic symptoms (axial hypertonia, opisthotonos, hypotonia or encephalitic symptoms). Myocarditis and prolonged tachycardia were identified in one case.

As far as gestational age at birth is concerned, it varied from 24 to 41 weeks. There was no difference between gestational age at birth in symptomatic and asymptomatic newborns (median = 34); however, stillbirth occurred in median at 32 weeks of gestation (*p*-value = 0.016).

We investigated the mode of delivery in livebirths, focusing on maternal and fetal indications for cesarian section (c-section): 6 (12.5%) neonates were delivered vaginally and 38 (77.08%) were delivered by c-section. In 4 cases, the mode of delivery was not reported.

Indications related to SARS-CoV-2 infection, such as severe maternal COVID-19 disease (3/38, 7.89%) and intrauterine fetal distress (20/38, 52.63%), were further investigated.

The 5-min Apgar Score varied from 2 to 10 (IQR:7–9.75). Although there was no difference between symptomatic and asymptomatic infected newborns in terms of Apgar Score (8.5 vs. 9, *p*-value = 0.22), in neonates extracted through c-section irrespective of the indication, extrauterine adaptation was more difficult as opposed to neonates born vaginally (9 vs. 10, *p* = 0.03).

### 3.2. Analyzing Primary Neonatal Adverse Outcome

Research was conducted to determine whether maternal clinical status could influence neonatal outcome. General and obstetrical metrics, as well as metrics related to SARS-CoV-2 infection and management of COVID-19 disease in pregnant patients, were stratified in function of neonatal outcome. (Table 4).

In terms of primary newborn outcome, neither maternal clinical status (symptomatic or not) at T1 nor the severity of COVID-19 disease made a difference; nevertheless, the presence of maternal fever and perception of diminished fetal movement demonstrated statistical relevance. There was a statistically significant difference between symptomatic and asymptomatic newborns when maternal fever was investigated (*p*-value = 0.037), as well as between livebirths and stillbirths when the perception of fetal movement was considered (*p*-value = 0.001) (Table 4).

Following the implementation of logistic regression analysis on the risk factors that were deemed statistically significant, our findings indicate that women who had fever during their SARS-CoV-2 infection were at a higher risk of giving birth to a symptomatic newborn (*p*-value = 0.016, OR: 4.55, 95% CI:1.33–15.57).

Nineteen out of 82 women reported reduced fetal movements at the moment of hospital admission. Almost 2/3 of the time, IUFD was discovered (*p*-value = 0.001). Overall, the presence of reduced fetal movement would increase the Odds Risk of stillbirth by 7.5 times (*p*-value = 0.001, 95% CI: 2.21–25.43). After excluding deliveries that occurred less than 28 weeks of gestation, the Odds Risk remained similar- 7.84 (*p*-value = 0.001, 95% CI: 2.23–27.5).

Of the studied obstetrical risk factors, only GA at T0 and T1 were statistically significant. There was a difference between livebirths and stillbirths concerning GA at the time of maternal infection and pregnancy outcome (Table 5).

A lower risk of stillbirth is associated with higher GA at T0 and T1 (*p*-value = 0.018, OR: 0.87, 95% CI: 0.79–0.97 for T0 respectively, *p*-value = 0.016, OR: 0.88, 95% CI: 0.78–0.98 for T1). Moreover, if maternal infection occurred after 30 weeks of gestation, the risk of stillbirth would decline almost by 2/3rds (*p*-value = 0.044, OR: 0.38, 95% CI:0.16–0.98).

In the current study, 9 out of 10 extremely preterm newborns had severe primary neonatal outcome. Most of them were stillborn (7/10); the rest had symptoms of COVID-19 disease (2/10). In this instance, the two symptomatic neonates died during hospitalization due to COVID-19 disease. The probability of an adverse neonatal outcome was still considerable for those born between 28 and 32 weeks of gestation (only 16% of them were asymptomatic). In the case of late preterm newborns, the risk of an unfavorable fetal outcome was still rather substantial (only 34% of the neonates were born without any symptoms). Less than half (42%) of the infants delivered beyond 37 weeks had an asymptomatic infection (Figure 3).

As such, we investigated whether the degree of prematurity makes a difference regarding primary neonatal outcome in livebirths. A Chi-Square test was used to compare the two nominal parameters, but no statistical significance was achieved (*p*-value = 0.26). (Table 6). Other neonatal metrics such as 5-min Apgar Score, birthweight, and presence of intrauterine growth restriction were stratified in live newborns and stillbirths (Table 6).

There was no statistical relationship between mode of delivery and pregnancy outcome (*p*-value = 0.66); however, newborns who were delivered through cesarean section had a significantly lower median gestational age at birth [34] in comparison to newborns who were delivered via vaginal delivery [37], (*p*-value = 0.036) (Figure 4). An increase in gestational age would lower the risk of cesarian section delivery in cases with congenital SARS-CoV-2 infection, irrespective of c-section indication (*p*-value= 0.046, OR: 0.73 95% CI: 0.54–1).

Nineteen neonates (39.58%) were delivered by c-section due to fetal distress, and 3 (6.25%) were delivered due to severe maternal COVID-19 disease. Most cases of intrauterine fetal distress occurred in pregnancies less than 37 weeks of gestation (16/19, 84.21%). Gestational age at the moment of maternal infection was statistically associated with intrauterine fetal distress. The higher GA at T0, the lower the chances of delivery due to intrauterine fetal distress (*p*-value = 0.017, OR: 0.84, 95% CI: 0.7–1.01).

Regarding neonatal clinical status (whether they were symptomatic or not), there was no statistical difference between neonates with intrauterine distress and those extracted for other indications (*p*-value = 0.43); however, newborns with intrauterine fetal distress were much smaller (Median = 2064) than those extracted due to other indication (Median = 2580) (*p* = 0.03).

### 3.3. Analyzing Secondary Neonatal Adverse Outcome

Secondary adverse neonatal outcomes were defined as neonatal death caused by COVID-19 disease and the existence of COVID-19 disease sequelae in the newborn. Out of the 48 livebirth cases, 3 deaths (6.25%) related to SARS-CoV-2 infection occurred during hospitalization and in the other 2 cases (4.16%), neurological symptoms persisted at the moment of discharge.

Prematurity (*p*-value = 0.03, OR: 0.64, 95% CI:0.43–0.96) was the only statistically significant indicator of neonatal death in case of congenital COVID-19 disease. All three newborns were delivered preterm with a median gestational age of 26 (Figure 5).

## 4. Discussions

The physiopathology of vertical transmission of viral infections can be determined by several variables. A virus has to cause maternal viremia, even for a brief time, in order to reach the placenta [95,96]. Viral strain and placental and fetal tropism of the virus are also crucial for vertical transmission [97]. Gestational age at the time of maternal infection may significantly influence the likelihood of congenital illness from two perspectives. Firstly, the placenta is a constantly developing organ, and its protective functions are at their peak in the latter stages of pregnancy. Secondly, the presence of high-affinity placental receptors may differ during pregnancy stages [98,99]. Eventually, the fetal immune response can make a difference if vertical transmission occurs. In this paper, we attempted to determine risk factors for adverse neonatal outcome in cases with congenital SARS-CoV-2 infection.

### 4.1. Maternal Metrics That Influence Neonatal Outcome

We compared maternal metrics to primary outcome to determine if maternal clinical state influences newborn outcome. The existence of neurologic-linked COVID-19 symptoms (anosmia, ageusia), maternal fever, maternal respiratory symptoms, flu-like symptoms, and the development of severe pneumonia were the investigated maternal risk factors. Of them, only the occurrence of fever was statistically significant. In symptomatic neonates with vertical transmission of the SARS-CoV-2, maternal fever was significantly more frequent (16/24) compared to asymptomatic neonates (8/24) (*p*-value = 0.03) (Table 1). In this study, the presence of fever in infected pregnant women increased the likelihood of newborn COVID-19 symptoms by 4.5 times (*p*-value = 0.016, 95% CI:1.33–15.57).

In order to pass the placental barrier, the presence of the virus in the bloodstream (viremia) is necessary. A recent matched controlled study concluded that low-level viremia is sufficient to infect the placenta [100]. Although viremia is associated with fever and flu-like symptoms in viral infections, the reverse is not a conditioned relationship. In an observational study that investigated viremia in 121 infected patients, fever was the most frequent symptom irrespective of the presence of the SARS-CoV-2 virus in the bloodstream (85.3% in patients with detectable viremia vs. 78.8% in patients without viremia) [101].The fever-related cytokine storm is another theory that might explain why neonates from feverish mothers had an increased likelihood of developing symptoms of infection, as an inframammary state of the fetus could weaken its immune system, allowing the manifestation of the disease [101].

In an attempt to identify other risk factors related to viremia, the presence of maternal severe COVID-19 pneumonia was investigated. Out of the 8 women diagnosed with severe disease, only 2 delivered asymptomatic neonates; the rest were either symptomatic (3) or were stillborn (3); however due to the small number of patients, statistical significance was not achieved (*p*-value = 0.88).

Of the obstetrical complaints, maternal perception of reduced fetal movements was the most frequent (19/82). Almost 2/3 of the time, intrauterine fetal death was diagnosed (*p*-value = 0.001). Multivariable analyses demonstrate that perception of reduced fetal movements is a strong predictor for stillbirth. Overall, the presence of reduced fetal movement in case of congenital SARS-CoV-2 infection increases the Odds of stillbirth 7.3 times (*p*-value = 0.001, 95% CI: 2.21–25.43) and 7.8 times in case of late stillbirths (*p*-value = 0.001, 95% CI: 2.23–27.5).

This is not surprising considering that fetal activity indicates a well-functioning central nervous system and periodic fetal movements have long been thought of as an indicator of fetal wellbeing [102]. Reduced fetal movement, on the other hand, has been associated with fetal distress and even intra-uterine fetal death; however, compared to normal population the Odds Risk is much higher in this study [103]. A three years case-control study, that was conducted before the pandemic (2006–2009) concluded that women with reduced fetal movement had a 2.37 risk of late stillbirth (95% CI: 1.29–4.35) [104]. A more recent study (2018) reports an Odds Risk of 4.51 (95% CI 2.38 to 8.55) [105]. When comparing the rate of stillbirth in infected SARS-CoV-2 women to those uninfected, an adjusted relative risk of 1.90 was reported ([aRR] = 1.90; 95% CI = 1.69–2.15) [26].

### 4.2. Obstetrical Metrics That Influence the Neonatal Outcome

The research was carried out to determine whether obstetrical factors like GA at T0, GA at T1 and ΔT could influence neonatal status. Gestational age at the moment of maternal diagnosis and at the moment of delivery played an important role as far as the outcome in livebirths is concerned. The higher the GA at the moment of maternal diagnosis and delivery, the lower the chances of stillbirths. One unit increase of GA at T0 will increase the Odds of stillbirth by 0.87 times (*p*-value = 0.018, 95% CI: 0.79–0.97), whereas one unit increase of gestational age at birth will increase the Odds of stillbirth by 0.88 times (*p*-value = 0.016, 95% CI: 0.78–0.98).

Extensive population studies have investigated the relationship between gestational age and the risk of stillbirth. All conclude that the risk is the highest in the second trimester (51% of all fetal fatalities occurred between the ages of 20 and 27 weeks) and gradually reduces with the increase of gestational age in pregnancies up to 37–38 weeks [106,107].

To better understand the link between prematurity and adverse primary outcome in pregnancies with vertical transmission of the coronavirus, all the cases were classified by degree of prematurity. Although the slope describing stillbirths decreases continuously across the three categories of prematurity, the frequency of stillbirth in this study remained high (70% in the extremely premature group and 34% in the moderate to late preterm group). Gestational age at the moment of infection and delivery seems to be very important in determining the primary and secondary outcomes of newborns with congenital SARS-CoV-2 disease. The higher the gestational age at the time of maternal infection/delivery, the lower the chances of the newborn showing no infection symptoms at birth. (OR: 1.16, 95% CI: 1.02–1.32 for gestational age at the time of maternal infection, OR:1.14,95% CI: 1.02–1.28, for gestational age at the time of delivery). Gestational age at the time of maternal infection seems to be a risk factor for severe intrauterine fetal distress that requires emergency c-section (*p*-value = 0.017, OR: 0.84, 95% CI: 0.7–1.01). While intrauterine fetal distress may be caused by placental or fetal components, severe placental dysfunction is the most plausible explanation in these circumstances, given that there is no statistical difference in the clinical outcome of neonates delivered due to fetal distress.

## 5. Strengths and Limitations

Although there have been numerous reviews on vertical transmission of the coronavirus, few use the standard criteria for proper vertical transmission diagnosis. Moreover, to our knowledge, this is the first research that analysis risk factors for adverse neonatal SARS-CoV-2 infection.

Nonetheless, there are some limitations to the present study. The study’s main limitation is that it comprises case reports and case series, despite using a systematic search strategy. According to the evidence-based medicine hierarchy, this research is at the lowest rank of the pyramid [108]. Including cases classified only as possible vertical transmission could be considered a limitation of the study, however, we counterbalanced by only including cases in which strict infection control and prevention procedures during delivery and mother-neonate separation for at least 24 h after birth were reported.

Also, despite the usage of rigorous inclusion criteria, some relevant cases aggregated in cohort studies may have been omitted owing to a lack of individual patient data. Despite following PRISMA-IPDrecommendations and other scoring methods that prevent bias, this issue remains a limitation of this research since it included only cases with congenital SARS-CoV-2 infection, an affliction that is considered to have a negligible rate of occurrence in the current literature [109].

## 6. Conclusions

Although congenital SARS-CoV-2 infection might be asymptomatic in certain instances, adverse neonatal outcomes are highly common. Maternal fever was a powerful predictor of symptoms occurrence in newborns (OR:4.5). Even after excluding severe preterm neonates, decreased fetal movement in pregnancies with in-utero transmission of the coronavirus was related to a high-risk IUFD (OR:7.8). The gestational age at the time of maternal infection and the time of birth both showed relevance in influencing the primary outcome in the analyzed lot.

Given the high publication bias, these findings should be interpreted with caution, and more extensive studies with proper diagnostic protocols should be conducted for confirmation. Nevertheless, this research can sensitives the medical community regarding the unseen peril of fetal coronavirus infection.

## Figures and Tables

**Figure 1 viruses-15-01615-f001:**
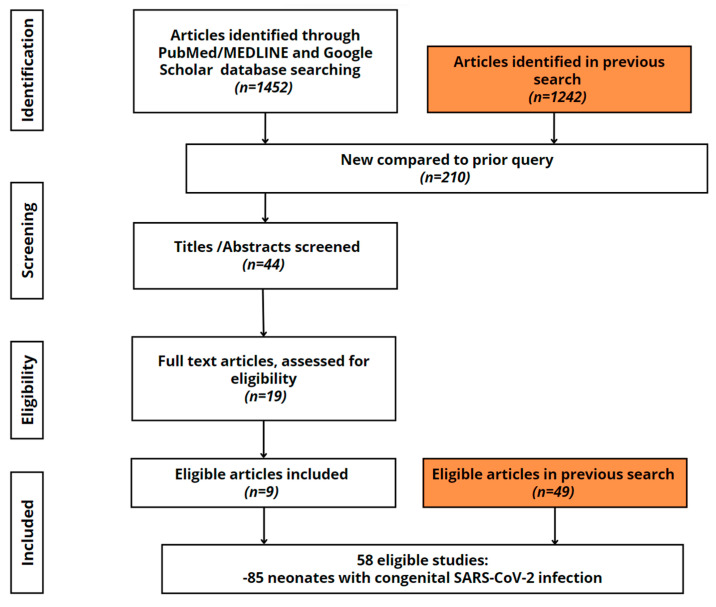
Individual participant data inclusion flow diagram. The orange boxes represent the findings of a prior scoping review [30]. In this case, three articles were excluded as birth occurred before 24 weeks of gestation. A total of 210 new articles were found, 44 of which were screened. Only nine publications remained suitable for the research once the inclusion criteria were applied. There were 58 studies in all, with 85 newbornswith confirmed or possible congenital SARS-CoV-2 infection.

**Figure 2 viruses-15-01615-f002:**
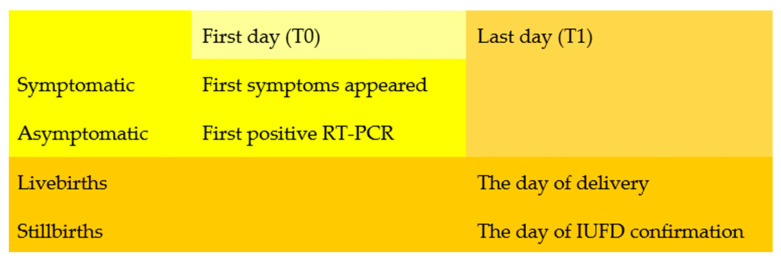
The detailed algorithm that presents the rationale when determining the ΔT.

**Figure 3 viruses-15-01615-f003:**
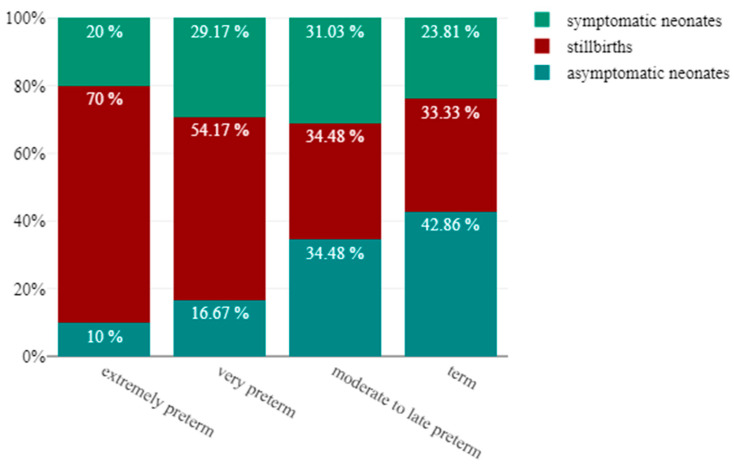
Degree of prematurity stratified by neonatal outcome (%). As GA advances, the rate of asymptomatic neonates rises (from 10% to 42.86%), and there is a decline in the stillbirth rate (from 70% to 33.33%). The rate of symptomatic neonates does not change drastically in the three categories of prematurity and term newborns (20% in the extremely preterm group, 29% in the very preterm group, 31.03% in the moderate to late preterm group and 23.81% in the neonates delivered at term).

**Figure 4 viruses-15-01615-f004:**
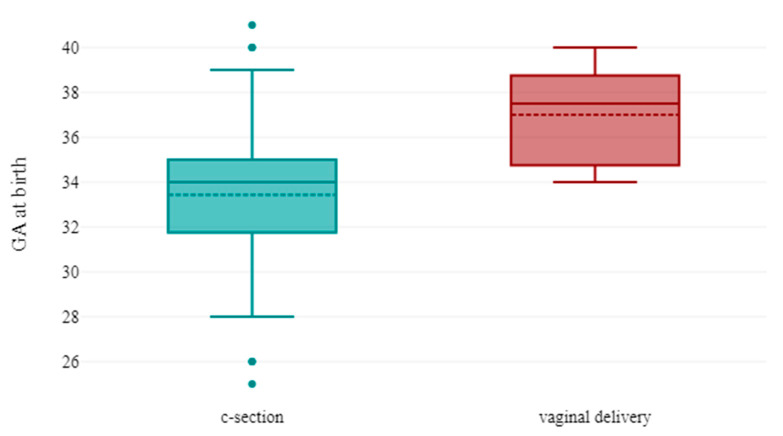
Boxplot represents mode of delivery in function of gestational age at birth. Premature newborns were more likely to be delivered by c-section.

**Figure 5 viruses-15-01615-f005:**
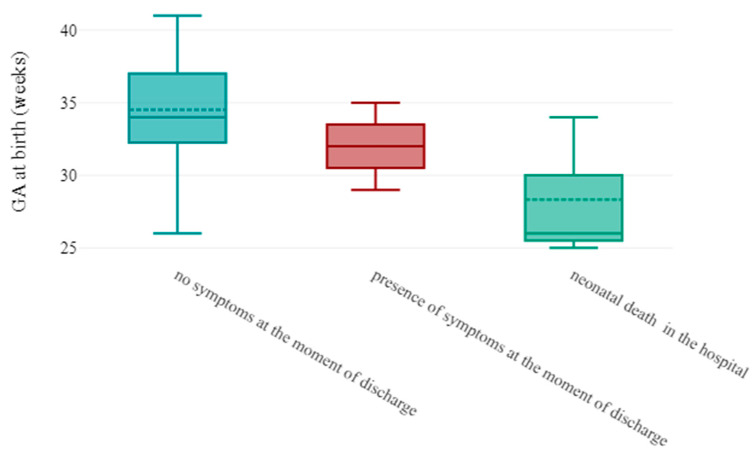
Boxplot represents secondary outcome in function of gestational age at birth. Extremely premature newborns were more likely to have fatal COVID-19 disease.

**Table 1 viruses-15-01615-t001:** Adapted algorithms of diagnosis in utero exposure in accordance with the WHO and Shah’s classification system [6,7]. IHC- immunohistochemistry.

Evidence of in-Utero Exposure			Evidence of Viral Persistence (24–48 h after Birth)	Type of Vertical Transmission
At Birth	<12 h after Birth	<24 h after Birth		
Positive RT-PCR from - sterile sample (neonatal blood, umbilical cord blood, tracheal aspiration, bronchial aspiration) OR - non-sterile samples (amniotic fluid, placental tissue, nasopharyngeal swab) OR - Positive IHC of the placenta			Positive RT-PCR from sterile sample	Confirmed intrauterine exposure according to the WHO classification system
			Positive RT-PCR from non-sterile sample OR positive serology	Possible intrauterine exposure according to the WHO classification system
Positive RT-PCR from - umbilical cord blood OR - neonatal blood OR - amniotic fluid (before rupture of membranes)		-	-	Confirmed intrauterine exposure according to the Shah classification system
Positive RT-PCR from - nasopharyngeal swab AND - placental swab (fetal side)	-		-	Possible intrauterine exposure according to the Shah classification system

**Table 2 viruses-15-01615-t002:** Adapted algorithms of diagnosis of intrapartum exposure following the WHO and Shah’s classification system [6,7].

Evidence of in-Utero Exposure (<24 h)	Evidence of Viral Persistence		Type of Vertical Transmission
	(24–48 h after Birth)	48 h–7 Days after Birth	
YES. At least one sampling performed that is negative.	Positive RT-PCR from sterile sample	-	Confirmed intrapartum exposure according to the WHO classification system
	Positive RT-PCR from non-sterile sample	Positive RT-PCR from non-sterile	
NO. No sampling was performed in the first 24 h of life.	Positive RT-PCR from sterile sample	-	Possible intrapartum exposure according to WHO classification system
	Positive RT-PCR from non-sterile sample	Positive RT-PCR from non-sterile	

**Table 3 viruses-15-01615-t003:** Adapted algorithms of diagnosis of intrapartum exposure following the WHO.

Fetal Tissue Sampling	Fetal Annexe Sampling	Type of Vertical Transmission
YES. Positive RT-PCR or ISH	-	Confirmed intrauterine exposure according to the WHO classification system
YES. Positive fetal swab or IHC	-	Possible intrauterine exposure according to the WHO classification system
NO	YES. Positive placenta (RT-PCR, ISH, swab) OR positive amniotic fluid	

**Table 4 viruses-15-01615-t004:** Distribution of potential maternal risk factors for adverse primary neonatal outcome in livebirths (asymptomatic, symptomatic newborns) and stillbirths diagnosed with congenital SARS-CoV-2 infection. All the missing values were excluded from the analysis. * Statistical significance: *p*-value < 0.05, ^(a)^ mean ± std. dev.; ^(b)^ Kruskal-Wallis statistical test; ^(c)^ observed frequency (percentage); ^(d)^ Chi-square statistical test (either asymptotic, Fisher’s exact test).

Maternal Metrics	Neonatal Outcome			*p*-Value ^(b),(c)^
	Livebirths		Stillbirths	
	Asymptomatic	Symptomatic		
Variables	N = 24	N = 24	N = 37	
Age ^(a),(b)^	30.78 ± 5.13	32 ± 5.62	30 ± 5.93	0.59
Symptomatic mother at the moment of delivery ^(a),(c)^	13 (54.16%)	19 (79.16%)	21 (56.75%)	0.18
Presence of neurologic-related COVID-19 symptoms ^(c),(d)^	2 (8.3%)	4 (16.66%)	4 (10.81%)	0.76
Presence of fever ^(c),(d)^	9 (37.5%)	17 (70.83%)	21 (56.75%)	0.13
	^(c)^ Asymptomatic vs. symptomatic, *p*-value = 0.037 *			
Presence of respiratory symptoms ^(c),(d)^	10 (41.65%)	16 (66.66%)	14 (37.83%)	0.81
Flu-like symptoms ^(c),(d)^	4 (16.66%)	3(12.5%)	6 (16.21%)	0.9
Presence of severe pneumonia ^(c),(d)^	2 (8.3%)	3 (12.5%)	3 (8.10%)	0.88
Comorbidities ^(c),(d)^	10 (41.65%)	11 (45.8%)	10 (27.02%)	0.26
Reduced fetal movement ^(c),(d)^	1 (4.16%)	3 (12.5%)	15 (40.54%)	0.002*
	^(c)^ Livebirths vs. stillbirths, *p*-value = 0.001 *			
Painful uterine contractions ^(c),(d)^	44 (16.66%)	4 (16.66%)	8 (21.26%)	0.96
Premature rupture of membranes ^(c),(d)^	11 (4.16%)	2 (8.3%)	-	0.5
Vaginal bleeding ^(c),(d)^	2 (8.3%)	1 (4.16%)	3 (8.10%)	0.85
Transaminitis ^(c),(d)^	11 (4.16%)	1 (4.16%)	1 (2.7%)	0.64
Thrombocytopenia ^(c),(d)^	2 (8.3%)	6 (25%)	7 (18.91%)	0.44
Antibiotic treatment ^(c),(d)^	3 (12.5%)	5 (20.83%)	2 (5.40%)	0.91
Antiviral treatment ^(c),(d)^	2 (8.3%)	4 (16.66%)	-	0.28
Anticoagulant treatment ^(c),(d)^	1 (4.16%)	1 (4.16%)	2 (5.40%)	0.43
Fetal lung maturation ^(c),(d)^	5 (20.83%)	4 (16.66%)	-	0.117
ICU admission ^(c),(d)^	11 (4.16%)	2 (8.3%)	1 (2.7%)	0.64
Invasive mechanical ventilation ^(c)^	-	2 (8.3%)	-	-

**Table 5 viruses-15-01615-t005:** Distribution of potential obstetrical risk factors for adverse primary outcome in livebirths (asymptomatic, symptomatic newborns) and stillbirths diagnosed with congenital SARS-CoV-2 infection. All the missing values were excluded from the analysis. * Statistical significance: *p*-value < 0.05, ^(a)^ median (Inter-Quartile Range, with Tukey’s hinges), ANOVA statistical test, ^(b)^ Mann-Whitney U-Test ^(c)^ observed frequency (percentage); Chi-square statistical test (either asymptotic, Fisher’s exact test).

Obstetrical Metrics	Neonatal Outcome			*p*-Value ^(b),(c)^
	Livebirths		Stillbirths	
	Asymptomatic	Symptomatic		
Variable ^(a),(b)^	N = 24	N = 24	N = 37	
GA at T0 ^(a)^	33 (32–38)	33.5 (29–35)	31 (27–34)	0.015 *
	^(b)^ Livebirths vs. stillbirths, *p*-value = 0.018 *			
GA at T1 ^(a)^	34	34	32	0.017 *
	34.83 ± 3.73	33.17 ± 4.13	31.17 ± 4.13	
	^(b)^ Livebirths vs. stillbirths, *p*-value = 0.016 *			
ΔT ^(a)^	7 (4.5–11.5)	7.5 (4.2–11.5)	10 (6–14)	0.33
c-section (overall) ^(b)^	21 (87.5%)	19 (79.16%)	-	0.66
c-section for intrauterine fetal distress ^(b)^	11 (45.8%)	8 (33.33%)	-	0.43
c-section for severe maternal COVID-19 disease ^(b)^	-	3 (12.5%)	-	-

**Table 6 viruses-15-01615-t006:** Distribution of potential neonatal risk factors for primary and secondary adverse neonatal outcome in livebirths (asymptomatic, symptomatic newborns) and stillbirths, diagnosed with congenital SARS-CoV-2 infection. There were missing values from the database, all the missing values were excluded from the analysis. ^(a)^ observed frequency (percentage); Chi-square statistical test; ^(b)^ median (Inter-Quartile Range, with Tukey’s hinges), Kruskal-Wallis statistical test.

Neonatal Metrics	Neonatal Outcome			*p*-Value ^(a),(b)^
	Livebirths		Stillbirths	
	Asymptomatic	Symptomatic		
Variable ^(a)^	N = 24	N = 24	N = 37	
Prematurity classification according to WHO [87]				
Preterm birth overall ^(a)^	15	18	30	0.23
Extremely preterm ^(a)^	1 (4.17%)	2 (8.7%)	7 (18.92%)	0.26
Very preterm ^(a)^	4 (16.67%)	7 (30.43%)	13 (35.14%)	
Moderate to late preterm ^(a)^	10 (41.67%)	9 (39.13%)	10 (27.03%)	
Term ^(a)^	9 (37.5%%)	5 (21.74%)	7 (18.92%)	
5 min Apgar Score ^(a)^	9 (7–10)	8.5 (7–9)	-	0.22
Birthweight (g) ^(b)^	1860 (1393–2441)	2267.5 (1575–2886)	2109.5 (1205–2765)	0.78
Distribution according to weight centile				
IUGR ^(a)^	2 (8.7%)	3 (12.5%)	2 (5.4%)	0.9
sIUGR ^(a)^	5 (21.74%)	2 (8.7%)	2 (5.4%)	0.2
Female newborns ^(a)^	9 (39.13%)	10 (41.66%)	17 (45.97%)	0.11
Neonatal death ^(a)^	-	2 (8.7%)	-	-
Neonates with secondary adverse outcome ^(a)^	-	3 (12.5%)	-	-

## Data Availability

Not applicable.

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
