# Peer review of "Maternal Fever and Reduced Fetal Movement as Predictive Risk Factors for Adverse Neonatal Outcome in Cases of Congenital SARS-CoV-2 Infection: A Meta-Analysis of Individual Participant Data from Case Reports and Case Series"

_viruses, 2023, doi:10.3390/v15071615_

Round 1

Reviewer 1 Report (Previous Reviewer 2)

no new comment

no new comment

Author Response

Point 1: Minor editing of English language required.           

Response 1: Thank you for your observation. The text was revised by a colleague fluent in English writing. Thank you for your effort to review the manuscript.

Reviewer 2 Report (New Reviewer)

Firstly, I would like to commend the authors on their efforts in undertaking this robust and insightful study. Overall, the manuscript provides valuable insights into the potential risk factors for adverse neonatal outcomes in cases of congenital SARS-CoV-2 infection, and the work presented is undoubtedly of high quality. 

A notable limitation of the study, which requires attention, is its sole reliance on case reports and case series. While the authors' effort in maintaining transparency and minimizing bias is commendable, I feel the title could benefit from a slight modification to more accurately reflect the scope of the study. The current phrasing, "individual participant data," while technically accurate, could potentially lead to confusion about the nature of the study's design and source material. 

Therefore, I would suggest a revision to the title that emphasizes the systematic review and meta-analysis of case reports and case series. This adjustment will accurately reflect the study design and enhance the understanding of the research. Given the opportunity, more assertive wording could certainly enhance the title's clarity and impact, improving its discoverability and appeal to the target audience.

A possible revised title could be: "Maternal Fever and Reduced Fetal Movement as Predictive Risk Factors for Adverse Neonatal Outcome in Cases of Congenital SARS-CoV-2 Infection: A Meta-Analysis of Individual Participant Data from Case Reports and Case Series."

I commend the authors for their efforts in undertaking this systematic review and meta-analysis, which no doubt contributes to our understanding of the issue at hand. I highly recommend this article for publication following this minor revision.

Author Response

Point 1: Firstly, I would like to commend the authors on their efforts in undertaking this robust and insightful study. Overall, the manuscript provides valuable insights into the potential risk factors for adverse neonatal outcomes in cases of congenital SARS-CoV-2 infection, and the work presented is undoubtedly of high quality.

A notable limitation of the study, which requires attention, is its sole reliance on case reports and case series. While the authors' effort in maintaining transparency and minimizing bias is commendable, I feel the title could benefit from a slight modification to more accurately reflect the scope of the study. The current phrasing, "individual participant data," while technically accurate, could potentially lead to confusion about the nature of the study's design and source material. 

Therefore, I would suggest a revision to the title that emphasizes the systematic review and meta-analysis of case reports and case series. This adjustment will accurately reflect the study design and enhance the understanding of the research. Given the opportunity, more assertive wording could certainly enhance the title's clarity and impact, improving its discoverability and appeal to the target audience.

A possible revised title could be: "Maternal Fever and Reduced Fetal Movement as Predictive Risk Factors for Adverse Neonatal Outcome in Cases of Congenital SARS-CoV-2 Infection: A Meta-Analysis of Individual Participant Data from Case Reports and Case Series."

Response 1: Thank you for the suggestions. The title was revised as per recommendation. A flowchart was added. We introduced details regarding the protocols used to diagnose vertical transmission. We complete details about SARS-CoV-2 manifestations in the newborn. We adjusted the design of tables 4, 5, and 6. We added more information about maternal vaccination status. We completed more details about SARS-CoV-2 manifestations in the newborn.  

Point 2: I commend the authors for their efforts in undertaking this systematic review and meta-analysis, which undoubtedly contributes to our understanding of the issue. I highly recommend this article for publication following this minor revision.

Response 2: We made the suggested revisions. Thank you for your appreciation and for your effort to review the manuscript.

This manuscript is a resubmission of an earlier submission. The following is a list of the peer review reports and author responses from that submission.

Round 1

Reviewer 1 Report

In this paper, the authors performed a meta-analysis and suggested that maternal fever and decreased fetal movement in cases of SARS-CoV-2 infection may be risk factors for poor neonatal prognosis. However, the paper makes no mention of maternal COVID-19 vaccination. Since maternal vaccination status is known to influence neonatal outcomes, the value of this study is very limited because maternal vaccination was not included in the analysis.

Table 1 is very confusing. The authors should include the number of Asymptomatic, Symptomatic, and Stillbirth neonates in the table. Also, it is not clear what the analysis of “Asymptomatic vs symptomatic” and “Livebirths vs stillbirths” in the table is and how it was done. Also, it is unclear how these results are described in the table.

none

Reviewer 2 Report

The analysis determine risk factors for primary and secondary adverse neonatal outcomes in newborns with congenital SARS-CoV-2 infection and found that maternal fever and reduced fetal movement may be predictive risk factors for adverse neonatal outcome in cases with congenital SARS-CoV-2 infection.

The reference of the definition of congenital SARS-CoV-2 infection is required. Such as the WHO criteria and NFSOG standards.

It is not easy to determine the neonatal adverse outcome origin: from congenital SARS-CoV-2 infection or from maternal SARS-CoV-2 infection.

In table 1, a total of 85 newborns with possible and confirmed congenital SARS-CoV-2 infection cases were concluded. The cases were not totally confirmed congenital SARS-CoV-2 infection! No newborns were negative controls?

In table 1, what does the “asymptomatic vs symptomatic“ meanThe case number does not match?

The Table 1 does not well design.

About the presence of fever and reduced fetal movement, the comparison is at the three positive group (asymptomatic, symptomatic and stillbirth). Is the adverse neonatal outcome mainly stillbirth? In the conclusions, “maternal fever and perception of reduced fetal movement may be predictive risk factors for adverse pregnancy outcome in cases with congenital SARS-CoV-2 infection”, what is the adverse pregnancy outcome mean?

Do have the data of SARS-CoV-2 genotype of the maternal infection?